# Anti-Inflammatory Activity of *Cnidoscolus aconitifolius* (Mill.) Ethyl Acetate Extract on Croton Oil-Induced Mouse Ear Edema



Eduardo Padilla-Camberos [1,*], Omar Ricardo Torres-Gonzalez [1], Ivan Moises Sanchez-Hernandez [1], Nestor Emmanuel Diaz-Martinez [1], Oscar Rene Hernandez-Perez [1] and Jose Miguel Flores-Fernandez [2,3,*]

[1] Unit of Medical and Pharmaceutical Biotechnology, Center for Research and Assistance in Technology and Design of the State of Jalisco, A.C. (CIATEJ), Guadalajara 44270, Mexico; polimerasados@gmail.com (O.R.T.-G.); isanchez@ciatej.mx (I.M.S.-H.); ediaz@ciatej.mx (N.E.D.-M.); operez@ciatej.mx (O.R.H.-P.)

[2] Centre for Prions and Protein Folding Diseases, Department of Biochemistry, University of Alberta, 204 Brain and Aging Research Building, Edmonton, AB T6G 2M8, Canada

[3] Department of Research and Innovation, Universidad Tecnológica de Oriental, Oriental 75020, Mexico

* Correspondence: epadilla@ciatej.mx (E.P.-C.); jose.flores@utdeoriental.edu.mx or floresfe@ualberta.ca (J.M.F.-F.); Tel.: +52-(33)-33455200 (ext. 1640) (E.P.-C.); +1-(825)-9931702 (J.M.F.-F.)

**Abstract:** Nowadays, there is a growing interest in the development of medicinal plant-based therapies to diminish the ravages of the inflammatory process related to diseases and tissue damage. Most therapeutic effects of these traditional medicinal plants are owed to their phenolic and antioxidant properties. *C. aconitifolius* is a traditional medicinal plant in Mexico. Previous characterization reports have stated its high nutritional and antioxidant components. The present study aimed to better understand the biological activity of *C. aconitifolius* in inflammation response. We developed an ethyl acetate extract of this plant to evaluate its anti-inflammatory capacity and its flavonoid content. The topical anti-inflammatory effect of the ethyl acetate extract of *C. aconitifolius* was determined by the croton oil-induced mouse ear edema test, while flavonoid detection and concentration were determined by thin layer chromatography and the aluminum chloride colorimetric assay, respectively. Topical application of the extract showed significant inhibition of the induced-ear edema (23.52 and 49.41% for 25 and 50 mg/kg dose, respectively). The extract also exhibited the presence of flavonoids. The finding of the anti-inflammatory activity exerted by the *C. aconitifolius* and the identification of its active principles may suggest and support its use for inflammation treatment.

**Keywords:** *Cnidoscolus aconitifolius*; inflammation; croton oil; flavonoids

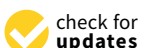

## 1. Introduction

Medicinal plants have been widely used for to treat several diseases since ancient times, and their application has spread throughout the world due to social diffusion. Nowadays, there are reports stating that medicinal plant extracts contain a large amount of phytochemical compounds responsible for their medicinal properties, such as terpenoids, essential oils, sterols, alkaloids, polysaccharides, tannins, anthocyanins, and phenolic compounds capable of stimulating the immmune system in disease. Currently, several studies are focused on plant extraction methods for preserving their biological properties and obtaining a good extraction yield [1–4].

In Mexico, there are two plant species known as Chaya: *Cnidoscolus chayamansa* and *Cnidoscolus aconitifolius*. The natural habitat of *C. aconitifolius* is in Mexico and Central America; it belongs to the family Euphorbiaceae and can grow on poor nutrient soils, besides being easily propagated and having a high resistance to pests and diseases [5].

Previously, it has been reported that both possess a high nutritional and antioxidant content, and owing to this, they are considered as a promising alternative to modulate hepatoprotection, lipid storage, insulin levels, pain, and inflammation [6–9].

Inflammation, considered as one of the principal responses of the immune system, involves redness, swelling, pain, heat, and dysfunction [10]. Non-steroidal anti-inflammatory drugs are used to treat inflammation. However, prolonged use may cause side effects. Hence, it is important to explore and develop new alternatives for the treatment of inflammation [11]. Previous studies have reported the physicochemical properties of *C. aconitifolius*. The chemical study carried out by Adeniran and Abimbade (2014) identified a new compound, 2,3-dimethoxy-5-vinylbenzene-1,4-dioic acid, and a known aromatic compound, 1,4-dimethylbenzene-1,4-dicarboxylate, in an ethanolic extract [12]. Godinez et al. (2019) identified for the first time eleven phenols in the species *C. aconitofolius* in aqueous and ethanolic extract [13]. Ajiboye et al. (2019) reported that the aqueous extract of *C. aconitifolius* showed the presence of total phenol and total flavonoids, in addition to determining its antioxidant activity [14]. On the other hand, García et al. (2014) demonstrated that the ethyl acetate extract from the leaf of *Cnidoscolus chayamansa*, a close relative of *C. aconitifolius*, has significant anti-inflammatory (with TPA-induced mouse ear edema) and cardioprotective activity (with ischemia/reperfusion (I/R) rat model), due to the presence of sterols, flavonoids, coumarins, and saponins [15]. The novelty of this work is the evaluation of biological activity in an animal model; therefore, this study aimed to determine the anti-inflamatory effect of an *C. aconitofolius* ethyl acetate extract and its total flavonoid content.

## 2. Materials and Methods

### 2.1. Plant Material

Fresh leaves of *C. aconitofolius* were collected from a local shrub located in Tonalá, Jalisco, Mexico, and the identification of the species was carried out by a specialist from the University of Guadalajara, Dr. Arturo Castro.

According to Jiménez-Aguilar and Grusak (2015), *C. aconitofolius* contains several minerals such as calcium, magnesium, potassium, phosphorus, sulfur, iron, sodium, and vitamin C [16]. Some physico-chemical characteristics of vegetable matter are: dark green color, pH of 1% solution 6.15 units.

### 2.2. Extract Preparation

Leaves were dried in an oven at 60 °C for 72 h, ground into fine powder using a mechanical grinder blender, and passed through an 850 μm sieve. For subsequent extraction, 15 g of powder were used in Soxhlet (Pyrex) at 77 °C using ethyl acetate as the solvent. This solvent is considered semipolar and has been used to extract both polar and non-polar compounds, and several studies have shown its efficacy in extracting polyphenols and flavonoids from plants [17]. The extracts were then concentrated in a rotary vacuum evaporator at 74 rpm and 45 °C. A final concentrate volume of 5 mL was recovered and stored at room temperature without light exposure. These operating conditions are suitable for the type of solvent used [18]. The extraction yield was calculated using the following equation:

$$\text{Yield of extraction}(\%) = \frac{\text{Weight of extract}}{\text{Dry weight of original sample}} \times 100 \quad (1)$$

where the dry weight of the original sample was 15 g.

### 2.3. Phytochemical Analysis

#### 2.3.1. Thin-Layer Chromatography

Thin-layer chromatography (TLC) was performed to confirm the presence of bioactive compounds as flavonoids; according to the United States Pharmacopeia [19], Quercetin (0.2 mg/mL) and Rutin (0.6 mg/mL) were used as standard controls. The TLC plate was developed using a mixture of ethyl acetate, water, glacial acetic acid, and formic acid (100:26:11:11), and test samples were diluted in methanol and water (4:1). A volume of 5 μL of each sample was applied to silica gel chromatoplates (Merck KGaA). Samples were

run separately, and results were observed under UV light. The Retention factor (Rf) was calculated by dividing the distance traveled by the compound, by the distance traveled by the solvent.

### 2.3.2. Total Flavonoid Assay

Total flavonoid content was determined by the aluminum chloride colorimetric method as reported by Marinova et al. (2005) [20] and Pekal (2014) [21] with slight modifications. The samples were analyzed in triplicates. One milliliter of the extract was added to 6.4 mL of distilled water, 0.3 mL of 5% $NaNO_2$, 0.3 mL of 10% $AlCl_3$, and 2 mL of 1M NaOH. The absorbance was measured at 510 nm, and the total flavonoid content of *C. aconitifolius* extract was expressed as mg of quercetin equivalents (QE)/mL. A calibration solution containing 0–400 mg/L of quercetin was used in distilled water. Samples were analyzed in triplicates.

### *2.4. Anti-Inflammatory Study*
### 2.4.1. Animals

Adult male Balb-C mice were housed under controlled temperature an illumination with food and water *ad libitum*. The experiments were conducted according to the guidelines established by the Federal Government of Mexico (NOM-062-ZOO-1999) [22], according to the "Guide for the Care and Use of Laboratory Animals" council for the National Institute of Health. All animal procedures were approved by the internal committee of CIATEJ who reviewed the protocol for the care of laboratory animals (approval number 2018-002-C).

### 2.4.2. Croton Oil-Induced Assay

Croton oil, indomethacin, and *C. aconitofolius* extract were dissolved in acetone and applied topically in the bundle and underside of the ear. All groups (*n* = 6) had croton oil applied in the right ear (1 mg/20 μL acetone), whereas the left ear received only acetone. Posterior to the croton oil application, indomethacin (10 mg/kg) was applied in the right ears, as well as *C. aconitifolius* extract (25 and 50 mg/kg) doses. After 4 h, mice were sacrificed to obtain ear biopsies (6 mm) to determine the anti-inflammatory activity by calculated weight differences between ears [23]. The control group did not receive any treatment. Data were expressed according to the following equation:

$$\%\text{Inhibition} = \frac{(\text{mean of ear biopsies weight}) \text{ control} - (\text{mean of ear biopsies weight}) \text{ samples}}{(\text{mean of ear biopsies weight}) \text{ control}} \times 100 \qquad (2)$$

### *2.5. Statistical Analysis*

Results are expressed as mean ± SEM. One-way ANOVA was used to compare differences between mean followed by Dunnett's test as a *post hoc* to compared treated groups versus the control group. Significant differences were considered when $p < 0.05$. Statistical analyses were computed using Prism (GraphPad Version 8) statistical software.

## 3. Results

### *3.1. Phytochemical Analysis*

The *C. aconitifolius* ethyl acetate extract analysis using TLC revealed a retention factor (Rf) of 0.79 and 0.93; these values were similar values obtained for the standard control Quercetin (Rf: 0.77 and 0.92) and different to values of standard control Rutin (Rf: 0.25 and 0.36) (Table 1).

**Table 1.** Retention factor values of *C. aconitifolius* extract and standard controls obtained during thin-layer chromatography (duplicate).

| Sample | Rf | Presence |
|---|---|---|
| *C. aconitifolius* | 0.79 | Q |
| | 0.93 | Q |
| Quercetin | 0.77 | Q |
| | 0.92 | Q |
| Rutin | 0.25 | R |
| | 0.36 | R |

Presence: Refer to similar values of Rf between extract and standard controls: Quercetin (Q) and Rutin (R).

In addition, TLC allowed to observe the colors of *C. aconitifolius* ethyl acetate extract and compared them to standard controls under UV light (Table 2 and Figure 1).

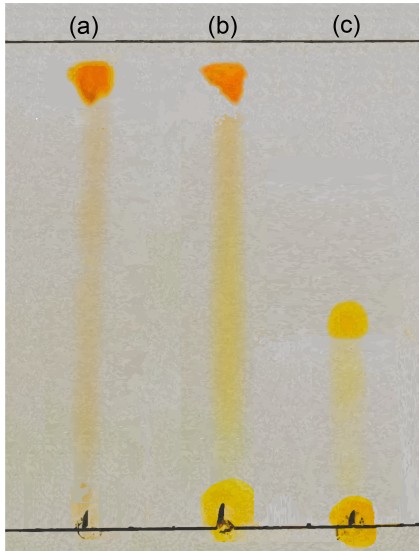

**Figure 1.** TLC results: (**a**) *C. aconitifolius* ethyl acetate extract, (**b**) Quercetin, (**c**) Rutin.

**Table 2.** Colorimetric determination of flavonoids observed in thin-layer chromatography under UV light.

| Sample | Color | Presence |
|---|---|---|
| *C. aconitifolius* | Orange | Q |
| Quercetin | Orange | Q |
| Rutin | Yellow | R |

Standard controls: Quercetin (Q) and Rutin (R).

Total flavonoids content of *C. aconitifolius* ethyl acetate extract was obtained using a standard calibration curve of quercetin solution as quercetin equivalents (mg QE/mL). *C. aconitifolius* contained 154.23 + 3.35 mg QE/mL as is shown in (Table 3).

**Table 3.** Total flavonoids determination of *C. aconitifolius* ethyl acetate extract.

| Quercetin (mg/mL) | Mean (mg/mL) |
|---|---|
| 154.58 | |
| 150.71 | $154.23 \pm 3.35$ |
| 157.39 | |

Results are expressed as mean ± standard error of mean (SEM). Values obtain to *C. aconitifolius* extract (10 mg/mL).

### 3.2. Induced Mouse Ear Edema with Croton Oil Assay

To test the anti-inflammatory effect of *C. aconitifolius* ethyl acetate extract, two doses of the extract (25 and 50 mg/kg) on mouse ear edema induced with croton oil were

applied. As is shown in Figure 2, the application of *C. aconitifolius* extracts on-ear induced a significant change in ear weight for both 25 mg/kg (one-way ANOVA; $p < 0.05$) and 50 mg/kg (one-way ANOVA; $p < 0.01$) doses in induced mouse ear edema assay compared to the control group. Indomethacin induced a significant decrease in the change in ear weight compared with the control group (one-way ANOVA; $p < 0.001$). The percent of inhibition of inflammation elicited by the *C. aconitifolius* ethyl acetate extract was 23.52% for a dose of 25 mg/kg and 49.41% for a 50 mg/kg dose. Indomethacin induced 87.05% of inhibition over inflammatory effect elicited by croton oil (Table 4).

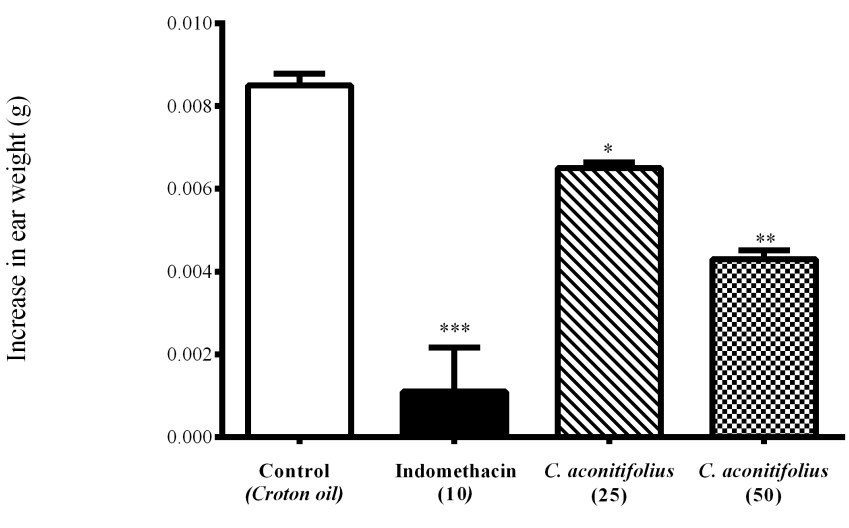

**Figure 2.** Anti-inflammatory effect by topical application of *C. aconitifolius* ethyl acetate extract on croton oil-induced ear edema. *C. aconitifolius* ethyl acetate extract in both doses decreased the ear edema. Indomethacin was used as positive control. Results are expressed as means ± SEM. N = 6. * $p < 0.05$; ** $p < 0.01$; *** $p < 0.001$. One-way ANOVA was followed by Dunnett's test.

**Table 4.** Ear weight differences and inhibition percentage in croton-oil-induced ear edema test.

| Treatment | Dose (mg/kg) | Change in Ear Weight (g) | Inhibition (%) |
|---|---|---|---|
| Control | - | 0.0085 ± 0.00028 | 0.00 |
| Indomethacin | 10 | 0.0011 ± 0.00106 *** | 87.05 |
| *C. aconitifolius* | 25 | 0.0065 ± 0.00014 * | 23.52 |
| *C. aconitifolius* | 50 | 0.0043 ± 0.00021 ** | 49.41 |

Results of change in weight are expressed as mean ± standard error of mean (SEM). N = 6 in each group * $p < 0.05$; ** $p < 0.01$; *** $p < 0.001$. One-way ANOVA. Control treatments refer to application of croton oil to induce acute inflammation.

## 4. Discussion

Topical application of the ethyl acetate extract of the *C. aconitifolius* leaf showed an anti-inflammatory biological activity in mouse ear edema induced by croton oil by decreasing the ear edema weight (Figure 2). In agreement with this, the percentage inhibition of the extract of *C. aconitifolius* ethyl acetate extract significantly inhibited the ear edema by 23.52% and 49.41% at 25 and 50 mg/kg doses, respectively (Table 4). The approach to determine flavonoid content confirmed the presence of quercetin in the *C. aconitifolius* ethyl acetate extract, which could be responsible for the decrease in inflammation. Flavonoids belong to the group of natural polyphenolic compounds with more than 4000 identified varieties. A large number of epidemiological, in vitro, and in vivo studies have documented the anti-inflammatory properties of a wide variety of flavonoids in different chronic inflammatory

conditions, such as autoimmune diseases, cancer, diabetes, cardiovascular disorders, and neurodegenerative diseases [24].

Our results suggest that *C. aconitifolius* extract has anti-inflammatory activity in the induced mouse ear edema by inhibiting the action of croton oil. A previous report describes the edema inhibition by topical application of *Cnidoscolus chayamansa* agents [7], another species of the genus *Cnidoscolus*. Similar results have been reported using an aqueous and ethanolic extract of *C. aconitifolius* with anti-inflammatory activity, decreasing the cytokines TNF-*α* and IL-6 by 46 and 48.38% in macrophages stimulated by lipopolysaccharide [25]. The in vitro anti-inflammatory activity exherted by *C. aconitifolius* ethyl acetate extract coincides with the effects reported on a carrageenan-induced paw edema of an *C. aconitifolius* ethanolic extract by percolation [1,26].

On other hand, the effect of indomethacin, a non-selective inhibitor of cyclooxygenase (COX) that reduces the production of prostaglandins, promoting pain and inflammation [27,28], as was expected, was the most effective to prevent mouse ear edema. Nevertheless, the outcome of this study will serve as the basis for further investigations in order to isolate bioactive compounds from *C. aconitifolius* ethyl acetate extract and test its anti-inflammatory properties in order to obtain a compound with similar activity to the indomethacin drug. The presence of quercetin flavonoids in *C. aconitifolius* ethyl acetate reported in this study and in the ethanolic extract was determined in the phytochemical characterization using TLC and aluminum chloride colorimetric assay [25]. Among flavonoids, quercetin has been described as the most abundant in vegetables, and their biological effects have been extensively studied [29]. Flavonoids target prostaglandins which are involved in the late phase of acute inflammation [30].

A relation between the presence of quercetin and the anti-inflammatory effect of *C. aconitifolius* is possible based on both in vitro demonstration of the inhibition of inflammation-producing enzymes COX [31] and in vivo amelioration of the inflammation induced by carrageenan [32].

## 5. Conclusions

The results of this work underline the role of *C. aconitifolius* extract on induced mouse ear edema with croton oil. It is disclosed that the application of 25 and 50 mg/kg of ethyl acetate extract of the *C. aconitifolius* showed anti-inflammatory activity, which can be explained by the quercetin extract that it contains. The obtained results of this study reveal the potential application of *C. aconitifolius* extract in the pharmaceutical field. However, further investigations are required regarding extraction process optimization, isolation and identification of bioactive compounds, and elucidation of cellular and molecular mechanisms in biological activity.

**Author Contributions:** E.P.-C., O.R.T.-G., N.E.D.-M.: Investigation, Conceptualization, Methodology, Writing— Review and Editing; I.M.S.-H.: Methodology, Software; J.M.F.-F., O.R.H.-P.: Data Curation, Formal Analysis, Resources; I.M.S.-H., O.R.T.-G., N.E.D.-M.: Data Curation and Visualization, Investigation, Formal Analysis; E.P.-C.: Investigation, Writing—Review and Editing, Acquisition of Financial Support for the Project, Resources, Management and Coordination Responsibility for the Research, Activity Planning and Execution. All authors have read and agreed to the published version of the manuscript.

**Funding:** This research received no external funding.

**Institutional Review Board Statement:** The study was an experimental trial approved by the internal committee of CIATEJ for animal health, approval number (2018-002-C).

**Informed Consent Statement:** Not applicable.

**Data Availability Statement:** The data underlying this article will be shared on reasonable request from the corresponding author.

**Conflicts of Interest:** The authors declare that there are no conflict of interest.

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
