# Peer review of "Anti-Inflammatory Activity of Cnidoscolus aconitifolius (Mill.) Ethyl Acetate Extract on Croton Oil-Induced Mouse Ear Edema"

_applsci, doi:10.3390/app11209697_

Round 1

Reviewer 1 Report

The manuscript “Anti-inflammatory activity of Cnidoscolus aconitifolius (Mill.) Ethyl acetate extract on croton oil-induced mouse ear edema” deals with the evaluation of the biological activity of C. aconitifolius in inflammation response. In particular, the bioactive compounds contained in this plant were extracted by ethyl acetate. The idea behind of the work is interesting; but, relevant revisions are required to make the work ready for publication.

Detailed comments:

- Abstract. Add quantitative results to this section.

- Introduction. The state of the art related to the use of plant extracts for medical application can be enlarged, as well as the performance of the different extraction techniques can be compared; for this purpose, see this recent review: Baldino et al., Supercritical fluid technologies applied to the extraction of compounds of industrial interest from Cannabis sativa L. and to their pharmaceutical formulations: A review, Journal of Supercritical Fluids, 2020, 165, 104960.

- The aim of the work is missing at the end of the Introduction. Please, add this part, highlighting the novelty of the work.

- M&M. Physico-chemical characteristics of the vegetable matter and of all the materials used have to be added to this section.

- Some typing errors are present. Please, check them carefully and correct.

- Extraction preparation. The operative conditions used have to be justified.

- R&D. A comparison of the obtained results with those found in the scientific literature has to be performed to underline the relevance of the present findings.

- Conclusion paragraph has to be enlarged adding a critical overview on the work.

- English can be improved.

Author Response

Response to Reviewer 1

-Abstract. Add quantitative results to this section.

Response: We add quantitative results in Abstract.

-Introduction. The state of the art related to the use of plant extracts for medical application can be enlarged as well as the well as the performance of the different extraction techniques can be compared for this purpose see the recent review: Baldino et al., Supercritical fluid technologies applied to the extraction of compounds of industrial interest from Cannabis sativa L. and to their pharmaceutical formulations. A review. Journal of Supercritical Fluids, 2020, 165. 104960.

Response: We revised the review suggested and other papers for enlarge the Introduction. Also, we include new references.

-The aim of the work is missing at the end of the introduction. Please add this part, highlighting the novelty of the work.

Response: We add the aim and novelty of the work at the end of the introduction.

-M&M. Physico-chemical characteristics of the vegetable matter and of all the materials used have to be added to this section

Response: Some physico-chemical characteristics of vegetable matter were added in Plant Material section.

-Some typing errors are present. Please check them carefully and correct

Response: We revised all text and make typing corrections about italics, sentence revised, and modified units.

-Extraction preparation. The operative conditions used have to be justified.

Response: Operative conditions in extract preparation were justified and added a new reference.

-R&D. A comparison of the obtained results with those found in the scientific literature has to be performed to underline the relevance of the present findings.

Response: Comparison of obtained results about anti-inflammatory activity of Cnidoscolus species are presented in references 7, 25, 1 and 26 of Discussion section.

-Conclusion paragraph has to be enlarged adding a critical overview on the work-

Response: Conclusion paragraph was revised and expanded.

-English can be improved

Response: We went through the entire manuscript to eliminate grammatical mistakes.

Reviewer 2 Report

The paper described the potential anti-inflammatory action of Chaya extract using croton oil-induced mouse ear edema model.

This work is a continuation of previous work of the Authors (Us-Medina et al., 2020) and provide new results. The results are interesting and deserve publication in my opinion.

I have only few recommendations before this work could be accepted for publication in Applied Sciences.

  • Justify the use of ethyl acetate as extraction solvent
  • Include TLC plate picture.

Few typing has to be corrected too, examples:

Line 54: « extraction.PNG »?

Before Figure 1: “Chaya 2.png”

Line 128: 4,000

Author Response

Response to Reviewer 2

-Justify the use of ethyl acetate as extraction solvent.

Response: The use of ethyl acetate was justified in Extract Preparation section (with a new reference).

-Include TLC plate picture.

Response: A picture of TLC plate is included but this is not very clear, I leave the inclusion of the photo to your consideration

-Few typing has to be corrected too. Examples: Line 54 “extraction PNG”. Before Figure 1 “Chaya 2 png”. Line 128: 4,000

Response: We revised all text and make typing corrections.

Some errors were caused by the change of format, since in the La Tex format they do not appear.

Round 2

Reviewer 1 Report

The authors answered all Reviewer's comments properly.